# Electrospun Graphene Nanosheet-Filled Poly(trimethylene terephthalate) Composite Fibers: Effects of the Graphene Nanosheet Content on Morphologies, Electrical Conductivity, Crystallization Behavior, and Mechanical Properties

**DOI:** 10.3390/polym11010164

**Published:** 2019-01-17

**Authors:** Chien-Lin Huang, Hsuan-Hua Wu, Yung-Ching Jeng, Wei-Zhi Liang

**Affiliations:** Department of Fiber and Composite Materials, Feng Chia University, Taichung 40724, Taiwan; janis781031@gmail.com (H.-H.W.); rattenfingero8@gmail.com (Y.-C.J.); bobby96124@gmail.com (W.-Z.L.)

**Keywords:** poly(trimethylene terephthalate), graphene, electrospinning, composite fiber, morphology, crystallization, electrical conductivity, mechanical property, elastic recovery

## Abstract

In this study the effects of increased graphene nanosheet (GNS) concentration on variations in the structure and properties of electrospun GNS-filled poly(trimethylene terephthalate) (PTT/GNS) composite fiber, such as its morphologies, crystallization behavior, mechanical properties, and electrical conductivity, were investigated. The effects of GNS addition on solution rheology and conductivity were also investigated. GNSs were embedded in the fibers and formed protrusions. The PTT cold crystallization rate of PTT/GNS composite fibers increased with the gradual addition of GNSs. A PTT mesomorphic phase was formed during electrospinning, and GNSs could induce the PTT mesomorphic phase significantly during PTT/GNS composite fiber electrospinning. The PTT/GNS composite fiber mats (CFMs) became ductile with the addition of GNSs. The elastic recoveries of the PTT/GNS CFMs with 170 °C annealing were better than those of the as-spun PTT/GNS CFMs. Percolation scaling laws were applied to the magnitude of conductivity to reveal the percolation network of electrospun PTT/GNS CFMs. The electrical conductivity mechanism of the PTT/GNS CFMs differed from that of the PTT/GNS composite films. Results showed that the porous structure of the PTT CFMs influenced the performance of the mats in terms of electrical conductivity.

## 1. Introduction

Graphene nanosheets (GNSs) that are several of nanometers thick and their related materials, such as carbon nanotubes (CNTs), are promising functional nanofillers for advanced applications because of their outstanding mechanical and electrical properties and large surface areas [1]. GNS-filled and CNT-filled polymer composites can be used in high-conductivity applications, such as electrostatic discharge devices [2], electromagnetic interference (EMI)-shielding materials [3,4], sensors [5], electrical switching [6], ambipolar field-effect transistors [7,8], and electrodes [9]. Furthermore, polymer/GNS composites are widely being explored because GNSs are affordable and effective alternatives to CNTs.

Electrospinning has attracted increasing attention in recent years as a method for fabricating polymeric fibers. Electrospun fiber mats exhibit considerable potential in energy devices, filtration, tissue engineering, and biosensors because of their high surface area-to-volume ratio, high porosity, and diverse nanostructures [10]. Furthermore, electrospun fibers filled with GNS or CNT can increase the functional properties of fibers for advanced applications. Su et al. [11] reported that the uniaxially oriented electrospun polyethylene oxide (PEO)/CNT composite fiber mats (CFMs) have better tensile strength, modulus, and electrical conductivity than randomly oriented electrospun PEO/CNT CFMs. Additionally, the crystallization rate and the perfection of PEO crystal in PEO/CNT fibers can be affected owing to the high extensional force and the bending jet yielded during PEO/CNT electrospinning. Kim et al. [12] found that although graphene oxide (GO) protrudes from the electrospun polyacrylonitrile (PAN) fibers, the carbon composite fiber mats (CFMs) with GNS retain good conductivity following PAN/GO fiber carbonization at 1000 °C. Li et al. [13] reported that the a highly sensitive H_2_O_2_ biosensor could be prepared via the electrospun poly(vinyl alcohol) (PVA)/GNS-decorated with silver nanoparticles CFMs. Jin et al. [14] reported that the electrospun PAN/CNT-cobalt ferrite CFMs exhibits an EMI shielding efficiency (SE) value of approximately 3.9 dB. Furthermore, Li et al. [15] successfully developed a novel poly(methyl methacrylate) (PMMA)/polyamide 6 (PA6)/GNS nanocomposite which was prepared from electrospun PA6/GNS and PMMA CFMs through hot press molding. The mechanical properties of the PMMA/PA6/GNS nanocomposites were significantly improved. Ramazani et al. [16] showed that the fiber diameter of polycaprolactone (PCL)/GO CFMs decreases as GO content increases because the viscosity of the PCL/GO solution decreases. GNS particles usually protrude when incorporated in electrospun fibers because the lateral dimension of the former is usually larger than the diameter of the latter. Therefore, GNS protrusions on electrospun composite fibers could be in contact with other substances. In our previous work [17], PVA solutions filled with GNSs were prepared for electrospinning, and the PVA/GNS CFMs showed good cell adhesion and proliferation, thereby suggesting their possible use in tissue engineering. This phenonmenon caused by the GNS protrusions could help the proliferation of cells. Gao et al. [18] showed that GNS could be anchored onto thermoplastic polyurethane (TPU) fibers through the combination of electrospinning and ultrasonication. Moreover, the conductivity of the TPU/GNS composite fibers significantly increased. However, limited information is available as to the electrical conductivity (*σ*) and GNSs dispersion state of polymer/GNS CFMs from electrospinning polymer solutions with various amounts of GNSs, especially with extra high GNS loading. Therefore, the novel composite fibers filled with GNS can be used in the development of functional objects for various of flexible applications, ranging from electrical sensors to tissue engineering.

The electrical property of composites filled with GNSs depends on the microstructural properties, such as GNS dispersion state, GNS–GNS interaction, and polymer–GNS interaction. The percolation scaling law is usually used as a theoretical basis for evaluating the microstructure of composites filled with fillers. The percolation equation is typically expressed as *P* ~ (ϕ
*−*
ϕ*_c_*)*^β^* [19,20], where *P* is the asymptotic behavior of a material property, ϕ is the filler volume fraction, *β* is the critical scaling exponent, and ϕ*_c_* is the fraction threshold. A filler network is formed that is higher than ϕ*_c_*, based on our previous study [21] where the percolated filler–filler network in composites in which the percolation exponent and threshold change was correlated with the filler geometric structure. Mazinani et al. [22] reported a polystyrene (PS) composite fiber filled with different contents and types of CNT. In addition, the *σ* of the electrospun PS/multi-walled CNT (MWCNT) CFMs was fitted using the percolation theory. Chien et al. [23] also demonstrated that the *σ* of the electrospun poly(d,l-lactic acid) (PDLLA)/carbon nanocapsule (CNC) CFMs could fit using the percolation theory. Sreeprasad et al. [5] demonstrated that polyallylamine hydrochloride (PAH) microfibers could be spun from a 40% PAH solution. The as-spun PAH fibers were immersed in a graphene quantum dot (GQD) solution and the covalently anchored GQDs on the PAH fibers were obtained to produce a percolating network to exhibit electron-tunneling transport for humidity and pressure sensor applications. However, to the best of our knowledge, the electrical percolation results of the electrospun polymer/GNS CFMs have not yet been reported.

Poly(trimethylene terephthalate) (PTT), a semi-crystalline thermoplastic polymer, is a member of the polyester family with an odd number of methylene groups. PTT possesses a faster crystallization rate and lower melting temperature than poly(ethylene terephthalate) (PET). Moreover, the PTT fiber also possess a high elastic recovery even at a high elongation of 20% [24]. Using PTT for an electrically conductive application has an advantage over that of other thermoplastic polymers, such as thermoplastic polyurethanes, because of its low water absorption, dimensional stability, better mechanical strength, and chemical resistance of the former [25,26,27]. PTT composites filled with nanofillers show enhanced properties and performance. Gupta et al. [28] demonstrated that PTT composites filled with 5 wt % MWCNT has an EMI SE value of approximately 20 dB. We previously reported that the PTT crystallization rate increases as GNSs are gradually added [29]. Meanwhile, in the case wherein liquid nitrogen was used to quench the melt, a PTT mesomorphic phase was formed despite the extremely short crystallization time with high GNS loading. The cold crystallization of the PTT composites was retarded by the GNSs. Moreover, the percolation exponent and threshold of the PTT/GNS composites are larger than those of the PTT/CNT composites [21]. In the electrospun PTT/CNT composite fibers, the aspect ratio and functionalized group of CNT affects the morphology of the resulting fibers [30]. However, the *σ*, morphologies, crystallization behavior, and mechanical properties of PTT/GNS composite fibers were poorly described, especially the elastic recovery of the electrospun PTT/GNS CFMs.

Khil et al. [31] reported that electrospun PTT fibers with diameters from 200–600 nm could be obtained via electrospinning with a trifluoroacetic acid (TFA)-methylene chloride co-solvent. Wang et al. [32,33,34] showed that electrospun PET, PTT, and polybutylene terephthalate could be prepared from a TFA solvent. Polyesters are soluble in strong polar solvents, such as TFA. However, TFA is not a good dispersion medium for GNSs. Therefore, determining the appropriate GNS dispersion in electrospun fibers remains a challenge. First, GNSs should demonstrate good dispersion in a semidilute polymer solution. Common processing routes for mixing polymers and fillers, such as melt compounding [35] and coagulation [36,37], were reported as a means to incorporate fillers into thermoplastic polymer matrices. Nevertheless, GNSs easily aggregate in polymer solutions with high GNS contents due to the van der Waals forces of attraction. Wu et al. [30] reported that PTT/CNT composites could be prepared via melt compounding for electrospinning. Moreover, PTT composites filled with GNS could be prepared through coagulation by using *ortho*-dichlorobenzene (*o*-DCB)-phenol as a solvent to obtain low electrical percolation thresholds of 0.53 and 0.84 vol % in GNS with different aspect ratios [21,29]. Coagulation should work better than melt compounding for GNS to be well dispersed and incorporated in thermoplastic polymer matrices because the low solution viscosity improves the polymer chain diffusion into the GNS interstices. Thus, fillers are dispersed in a polymer matrix to form composite powders, and the composite powders are re-dissolved in a solvent to obtain a well-dispersed polymer/filler solution. This method may be an appropriate route for preparing polyester/GNS solutions for electrospinning.

Although many achievements have been reported in this area, the polymer chain conformations and crystal structures in semicrystalline polymer/GNS composite nanofibers are incompletely known. Thus, understanding how GNS contents interact with electrospun composite fibers and how the functional properties of the fibers can be controlled for advanced applications, such as mechanical and electrical applications is of great interest. In this study, PTT composite powders filled with well-dispersed GNSs were prepared via coagulation, and the composite powders were re-dissolved in a TFA solvent to obtain a well-dispersed PTT/GNS solution. To clarify the electrified jet manipulation for producing PTT composite fibers filled with GNS, systematically studying the effects of GNS on solution properties and the electrospinning process is necessary. Thus, the rheological and conductive properties of the prepared PTT/GNS solution were measured and correlated with the electrospinnability characteristics of the solution. The morphologies of GNS assemblies in the PTT fiber with extra high GNS loadings were also investigated. GNSs formed protrusions on the PTT/GNS composite fibers, and these protrusions helped the formation of the GNS–GNS network in the PTT/GNS CFMs. The influence of GNS loading on the cold crystallization of PTT/GNS CFMs was analyzed via differential scanning calorimetry (DSC), Fourier transform infrared spectroscopy (FTIR), and wide-angle X-ray diffraction (WAXD). GNSs could induce the PTT mesomorphic phase in PTT/GNS composite fibers significantly during PTT/GNS electrospinning. Moreover, the *σ*, cold crystallization behavior, and mechanical properties of the PTT/GNS CFMs were studied to reveal the GNS dispersion state, GNS–GNS network, and microstructure in the electrospun PTT composite fibers. The advantages for incorporation of GNS in the electrospun PTT fibers increased the electrical conductivity and ductility of the PTT/GNS CFMs.

## 2. Materials and Methods

### 2.1. Materials and Composite Preparation

PTT was purchased from DuPont Co. (Wilmington, DE, USA), and the viscosity-average molecular weight of PTT was determined to be 53,100 g/mol [29]. *o*-DCB was purchased from Acros Organics (Morris Plains, NJ, USA). Phenol was purchased from Showa Chemical Co. (Tokyo, Japan). GNSs were purchased from Enerage Inc. (P-LF10, Yilan, Taiwan), which were prepared via thermal reduction of GO at 1500 °C. According to the manufacturer, the average thickness of the GNS was 3 nm. The PTT and GNSs had densities of 1.28 and 2.20 g/cm^3^ [21], respectively. The GNSs were added to the *o*-DCB-phenol (1:1, *v*/*v*) solvent, after which ultrasonic treatment was performed for 6 h. Weighed PTT pellets were then added. Afterward, the mixture, with a 1% (*w*/*v*) solid content was stirred at 140 °C to dissolve the PTT pellets. The uniform suspension solution was precipitated dropwise into a 20-fold excess volume of methanol. The precipitated powders were continuously dried in a vacuum oven at 120 °C for 72 h until the residual solvent was removed. Note that the absence of any *o*-DCB absorbance peaks at 662 cm^−1^ of the FTIR spectra, which were assigned to C–Cl stretching vibration, and phenol peaks at 810, 1160, and 3421 cm^−1^, which were attributed to the asymmetric stretch of phenolic C–C–OH, C–O, and O–H stretching vibration, respectively, indicated the complete removal of residual solvent [21]. 

### 2.2. Poly(trimethylene terephthalate)/Graphene Nanosheet (PTT/GNS) Solution Preparation and Properties

To prepare a homogenous PTT/GNS solution with varying GNS amounts (based on the PTT polymer), pre-weighed PTT powders were added to a TFA solvent and vigorously stirred for several hours. A PTT solution (14 wt %) was used to monitor the effect of carbon nanofillers on the morphology of the as-spun fibers. A PTT solution (14 wt %) filled with different amounts of GNSs (based on the weight of the PTT polymer) was then obtained [17]. In the present study, the samples were designated according to their polymer-to-filler weight ratios, for example, 99/1. The corresponding volume fraction ϕ was obtained from the respective densities of the applied components. The solution conductivity properties (*κ*) were measured at 25 °C using a Consort conductivity meter (C832, Consort, Turnhout, Belgium). The viscosities of the solutions were measured using a viscometer (DV-II+Pro, spindle 18, and cup 13R, AMETEK Brookfield, Middleboro, MA, USA) at 25 °C. The viscoelastic pr−operties of the solutions filled with GNSs were measured in a strain-controlled rheometer (DHR-1, TA Instruments, New Castle, DE, USA) using a flat plate feature with a 60 mm diameter at 25 °C.

### 2.3. Electrospinning Process

The prepared solutions were subjected to room temperature electrospinning, wherein the nozzle size was *D_i_*/*D*_0_/length = 0.69 mm/1.09 mm/4 cm, and *D_i_* and *D*_0_ denoted the inner and outer nozzle diameters, respectively. The prepared solutions were delivered by a syringe pump (Cole–Parmer, Vernon Hills, IL, USA) to the nozzle at a controlled flow rate (*Q*). A high electrical voltage (*V*) was applied to the spinneret using a high-voltage source (MECC, HVU-40P100, Fukuoka, Japan) to provide a sufficient electric field for electrospinning. To construct a needle-to-plate electrode configuration, an aluminum board (30 × 30 cm^2^) was used as the collector for the electrospun fibers at a fixed tip-to-collector distance (*H*) of 14 cm. A summary of the different carbon nanofillers and concentrations studied here and the resulting fiber morphologies is shown in Table 1.

### 2.4. Morphology and Characterization of Fibers

The morphology of the fibers was observed using a scanning electron microscope (SEM, Hitachi S4100, Krefeld, Germany). Fiber diameters were measured from a collection of approximately 200 fibers, from which the average diameter (*d_f_*) was determined. A transmission electron microscope (TEM, Jeol JEM-1200EX, Peabody, MA, USA) was used to determine the locations of the carbon nanofiller particles within the fibers. These PTT/GNS CFMs were subsequently evaluated on the basis of thermo-gravimetric analysis (Appendix A). There was no significant weight loss before degradation. Thus, no residual solvent was detected in the PTT/GNS CFMs. FTIR measurements were performed using a Perkin-Elmer FTIR spectrometer (Spectrum Two, Waltham, MA, USA) equipped with a Mettler heating stage (HT82) for temperature control. A total of 32 scans with a 2 cm^−1^ resolution were obtained for each spectrum with a holding time of 8 min. The wide-angle X-ray diffraction (WAXD) intensity profiles of the PTT/GNS CFMs were obtained using a Bruker diffractometer (NanoSTAR Universal System, Cu *K_α_* radiation, Billerica, MA, USA). In-situ intensity profiles of samples during heating were obtained using a vacuum-assisted heating device. The heating process was realized by a stepwise increase of temperature with a holding time of 20 min each for data acquisition. The crystallization and melting behavior of the PTT/GNS CFMs was investigated using a TA differential scanning calorimetry (DSC) Q20 under a nitrogen atmosphere. The samples were heated to 270 °C at a rate of 10 °C/min.

The PTT/GNS composite films were prepared by hot pressing the PTT/GNS CFMs into rectangular steel molds (with a thickness of 0.3 mm) at 280 °C for 3 min, followed by air cooling to room temperature. Measurements for the samples with high *σ* (>10^−6^ S/cm) were performed using a Keithley 2400 Source Meter (Solon, OH, USA). A standard four-probe technique with an applied voltage of 1–20 V was applied to reduce the effects of contact resistance. A Keithley 6487 electrometer (Solon, OH, USA) equipped with a Keithley 8009 resistivity fixture was used for the samples with low *σ* (<10^−6^ S/cm) based on the ASTM D257 standard. The applied voltage for the sample with low *σ* was between 100–500 V. 

The mechanical properties of the as-spun PTT/GNS CFMs with and without annealing were obtained through tensile strength testing. During annealing, the PTT/GNS CFMs were heated up to 170 °C for 30 min in a vacuum and without any stretching. For example, a sample of PTT/GNS 99/1 CFMs with 170 °C annealing is denoted as 99/1@170. The mechanical properties of the PTT/GNS CFMs were obtained from the stress-strain curve using a universal tensile testing machine (HT-2402, Hung Ta Intsrument Co., Taichung, Taiwan) at a stretching rate of 10 mm/min. Specimens with a dog-bone shape were prepared using a sharp cutter. The mean thickness of each sample was ~50 μm. The gauge length and sample width were 15 and 4 mm respectively. The reported data of mechanical properties represent the average results of the five tests. The sample is drawn 10 mm/min to 20% of its initial gauge length. Thereafter, the tensile load was removed immediately, and the crosshead was allowed to returned to its original position with the same crosshead speed. These measurement protocols were repeated five times. The elastic recovery for the first and five stretch cycle was used in the following equation [24]:(1)$$\mathrm{elastic}\;\mathrm{recovery}=\frac{L_{o}-L}{L_{o}}\times 100\%$$
where *L_o_* is the original length of the filament and *L* is the irreversible length of the filament after the crosshead it was returned to its original position.

## 3. Results and Discussion

### 3.1. Effect of Carbon Nanofillers on Solution Properties

In general, obtaining bead-free fibers is closely correlated to a polymer’s solution concentration and molecular weight. However, solvent volatility affects the electrospun fiber morphology. Khil et al. [31] demonstrated that smooth PTT fibers could be electrospun via a 13 wt % PTT/TFA-methylene chloride (MC) solution (TFA-MC 1:1, *v*/*v*). Wu et al. [30] showed electrospun PTT fibers using different ratios of TFA/MC solvent. As the MC content decreased, a coalescent fiber structure was observed because of the reduced volatile MC content. Wang et al. [32] reported that a smooth PTT fiber could be electrospun using neat PTT/TFA solution. Therefore, TFA is an excellent solvent in polyester electrospinning for producing fine diameters. Based on the method by McKee et al. [38], a log–log plot of the solution’s specific viscosity (*η_sp_*) versus the volume fraction (ϕ_PTT/TFA_) was constructed to determine the *C_e_* (obtaining a semidilute solution regime with entangled PTT chains) for the present PTT/TFA solutions. Figure 1 shows the plots of the *η_sp_* versus the volume percentages of the investigated PTT solutions. The *η_sp_* of the PTT/TFA solution increased with the increased PTT content. The rapid viscosity increase at 9.44 vol % (8 wt %) suggests that *C_e_* occurred approximately at this concentration, and *η_sp_* and the volume percentage of the PTT/TFA solution had a constant slope of 3.93 above this concentration. The determined exponent is consistent with the theoretical prediction for entangled solutions in a good solvent (~3.9). The overlapping concentration *C** of 0.91 vol % was determined by 1/[*η*]. Therefore, the *C_e_*/*C** ratio is ~10.4, which is in agreement with that reported ratio for polymers dissolved in good solvents. Moreover, the measured *κ* for the neat TFA solvent is 1.26 μS/cm. A PTT concentration increase to 14 wt % results in slight increase of the *κ* for 14 wt % PTT/TFA solution to 8.31 μS/cm.

Appendix A shows the SEM images of the PTT fiber products collected from the electrospinning of the PTT solutions with different concentrations. The minimum concentration required for the PTT solutions to produce bead-free fibers is 9 wt %. PTT solutions with concentrations higher than 9 wt % possess a stretchable PTT chain network that prevents network fracture during electrospinning, thereby yielding bead-free fibers. In general, electrospinning eventually degenerates to electrospraying in solutions with a concentration lower than the entanglement concentration (*C_e_*). To produce uniform fibers via electrospinning, the solution concentration should be as high as 1.8~2.5 *C_e_* [38,39]. McKee et al. [38] reported that the minimum solution concentration required to electrospin bead-free fibers is 2.0~2.5 *C_e_* for a PET copolymer/chloroform-dimethylformamide solution. However, for the present PTT/TFA solution, the minimum concentration required to obtain smooth fibers is 9 wt %, which is similar to the *C_e_* of 9 wt %. This result revealed that the minimum concentration for producing smooth PTT fibers is approximately 1.0 *C_e_*, which could be attributed to the high TFA volatility. Wang et al. [32,33] reported similar results. Therefore, we selected 14 wt % PTT solution (~1.5 *C_e_*) filled with different amounts of GNS to ensure a completely stable fiber formation during electrospinning and to prevent non-smooth fiber morphologies of PTT/GNS composite fibers. Non-smooth morphologies were identified as bead fibers resulting from the weak network of a PTT solution.

Figure 2 shows the structures of the deposited GNS on the TEM grid prepared from the *o*-DCB solution. GNSs were transparent with folding and wrinkling. In the TEM observation in our previous study [21], the number-average long and short axis lengths of GNS were 1.67 ± 1.48 and 0.65 ± 0.49 μm, respectively. Moreover, the number-average minimum thickness of GNS based on atomic force microscopy (AFM) measurements was 6.5 nm. Numerous studies have incorporated GNSs into polyester via melt compounding; however, fully penetrating the GNS interstices is difficult for the polymer chains of matrices. Based on the above PTT/TFA solution results, TFA is a good solvent for a PTT polymer. Therefore, the PTT/GNS composite powders were prepared via coagulation. Consequently, the PTT/GNS composite powders were re-dissolved in the TFA solvent to prepare the PTT/TFA solution with different amounts of GNS for electrospinning.

A 14 wt % PTT solution filled with various amounts of GNS can be used to obtain PTT/GNS composite fibers via electrospinning. However, the polymer or polymer solution filled with carbon nanofillers could form nanofiller-nanofiller networks [23,40,41]. Thus, to determine the GNS–GNS network effect of GNS addition on a PTT solution, viscoelastic measurements of the 14 wt % PTT solutions filled with 1–7 wt % GNS were conducted. Figure 2a,b show the *G′* and *G″*, respectively, of the 14 wt % PTT solutions filled with different GNS amounts. A slight GNS addition of 1 wt % did not change the *G′* in the low frequency region. When the GNS contents were increased to 3 wt %, the slope in the low frequency region was reduced and a small plateau was observed; however, the increase in GNS content had no effect in the high frequency region. For the PTT solution filled with 5 wt % GNS, a two-order increase of *G′* in the low frequency region was observed and eventually shifted the whole curve upward. The large plateau increase at the low frequency region indicated a GNS–GNS network formation in the entangled polymer chain network. A mild increase in *G″* was observed with the corresponding GNS addition. This finding suggests an increased elasticity of the solution. The rheological properties of the GNS-filled PTT solutions were significantly changed; therefore, their influence on electrospinnability deserves further discussion.

Figure 3c shows the complex viscosity (*η**) of the PTT solutions filled with different GNS amounts at 25 °C. Newtonian flow behavior occurred in the neat PTT solution during measurement frequencies, and the value of the zero-shear viscosity (*η_o_*) curves reached 0.88 Pa-S. The *η** value was unchanged when 1 wt % GNS was added to the solution. When the GNS content was increased to 5 wt %, Newtonian flow behavior occurred at low frequencies, but the behavior became non-Newtonian at frequencies higher than 0.75 rad/s. The *η** of the PTT solution with 7 wt % GNS was significantly increased, and *η_o_* reached 2.57 Pa-S. Ramazani et al. [16] examined the viscosity of a PCL/GO solution and concluded that *η** decreases as the GO content increases due to the induced free volume around the GO nanosheets. Our viscosity result showed that *η** increased as the GNS content increased in the PTT/GNS solution, which indicates the formation of a GNS–GNS network, and the absence of a free volume effect around the GNS. Appendix A shows the conductivity of the PTT solution filled with GNS. GNSs are conductive fillers; therefore, adding GNS into PTT solutions significantly improves the *κ* values. After adding 7 wt % GNS to the PTT solution, the *κ* reached 151.1 μS/cm, which is significantly higher (18-fold) than that of the neat PTT solution. This result indicates that adding GNS to PTT solutions could significantly increase the PTT solution conductivity. Thus, the addition of GNS increased the viscosity and conductivity of the PTT solution, and a GNS–GNS network was formed in the 14 wt % PTT solution filled with 5 wt % GNS.

### 3.2. Effect of GNS Concentration on Electrospinning and As-Spun Fiber Morphology

Figure 4 shows the functioning domain for electrospinning a 14 wt % PTT solution with various GNS contents. Functioning domains [23,42,43] are defined as the operating windows of applied voltage and *Q* required for a stable cone-jet mode. The lower- and upper-bound applied voltages are denoted as *V_s_* and *V_us_*, respectively. An *H* of 14 cm was used given the volatility of the TFA solvent. At a given *Q*, the operating windows (*V_us_* − *V_s_*) were higher for the PTT solution filled with GNS content than those for the neat PTT solution. Moreover, increasing the GNS content increased the operating window at a given *Q*. Based on the functioning domain for electrospinning PTT/GNS solutions (Figure 4), a common but limited processing window does exist to determine the GNS effect. Therefore, *Q* and *V* were determined to be 0.3 mL/h and 11 kV, respectively, in electrospinning the PTT/GNS solution to demonstrate the GNS effects on fiber diameter.

To determine the effect of GNS additions on fiber morphology and diameter, 14 wt % PTT solutions filled with 1–7 wt % GNSs were electrospun and compared. Figure 5 shows the SEM images of the fibers collected from electrospinning 14 wt % PTT solutions with various GNS amounts with the following conditions: *Q* = 0.3 mL/h, *H* = 14 cm, and *V* = 11 kV. The thick arrows indicate the GNS positions. Based on the SEM micrographs shown in Figure 5, the PTT composite fibers became less smooth and formed an increasingly irregular structure along the fiber as the amount of GNS increased. At 5 wt % GNS content, the PTT/GNS composite fibers with nanofibrils became observable. The thin arrows indicate the positions of the nanofibril whose diameters are 10–30 nm.

The TEM images in Figure 5 show that GNSs are identifiable and that they are embedded in the PTT fibers. The lateral dimension of GNS was larger than the diameter of the electrospun PTT fibers, and the GNS particles protruded from the smooth PTT fiber. At an increased GNS content in the PTT solution, numerous GNS protrusions were observed on the fiber surface, with some GNS appearing curly in the PTT/GNS composite fibers. The GNS protrusions could help the formation of the GNS–GNS network in the PTT/GNS CFMs. The details of this observation will be discussed later. At 7 wt % GNS content, GNSs became distinctly distributed in the PTT fiber (Figure 5h), and the inter-GNS distance decreased *(L_GNS_)*.

The PTT/GNS composite fiber diameters were measured from a collection of over 200 fibers under the same electrospinning conditions (*Q* = 0.3 mL/h, *H* = 14 cm, and *V* = 11 kV). For a neat PTT fiber, the measured *d_f_* was 256 ± 92 nm. At 1 wt % GNS content, the *d_f_* of the PTT/GNS 99/1 fiber decreased significantly to 141 ± 28 nm. When the GNS content reached 3 wt %, the *d_f_* of the PTT/GNS 97/3 fiber increased to 310 ± 116 nm. Furthermore, when the GNS content was increased to 5 and 7 wt %, the *d_f_* of the PTT/GNS 95/5 and 93/7 fibers slightly increased to 359 ± 133 and 364 ± 114 nm, respectively. The *d_f_* of the PTT/GNS fiber initially decreased and then increased as the GNS content increased. The nanofibrils in the PTT/GNS CFMs increased the standard deviation of *d_f_*. In electrospinning, solution viscosity and conductivity are important factors for determining the *d_f_* of electrospun fibers. In previous studies [44,45], *d_f_* was shown to decrease with a decreasing solution viscosity and increasing solution conductivity. Based on the results shown in Appendix A, the conductivity of the PTT/GNS solutions increased as the GNS content increased. High solution conductivity mainly causes the decrease in the *d_f_* of PTT/GNS composite fibers because it increases the electrostatic force, thereby stretching it during electrospinning. Based on the results shown in Figure 2c, the solution viscosity of the PTT/GNS solution filled with 1 wt % GNS was almost the same as that of the neat PTT solution. However, the conductivity of the PTT/GNS solutions filled with 1 wt % GNS was increase significantly, which resulted in the higher stretched forced in an electrostatic field. This phenomenon resulted in a decrease in the *d_f_* of the PTT/GNS 99/1 composite fiber. When the GNS content exceeded 3 wt %, the solution viscosity of the PTT/GNS solution increased, which resulted in the higher flow retardation during electrospinning. This phenomenon resulted in an increase in the *d_f_* of the PTT/GNS composite fiber when the GNS content was further increased. Therefore, the initial change in *d_f_* was attributed to a greater increase in solution conductivity than in solution viscosity, whereas the final change in *d_f_* was attributed to a greater increase in solution viscosity than in solution conductivity. This finding is in agreement with PET/CNT [46], PDLLA/CNC [23], and PVA/GNS [17] composite fibers.

The increased GNS content to 10 wt % in the 14 wt % PTT solution results in the viscosity of PTT/GNS solution to become too high to manipulate a stable cone-jet mode. Eventually, the PTT/GNS solution was dried at the nozzle end during electrospinning. Non-Newtonian flow behavior occurred for the 14 wt % PTT solution filled with 10 wt % GNS during the measurement of the shear rate, and *η* reached 11.52 Pa-S at 8.4 1/s (Appendix A). Therefore, the PTT solution with high GNS content must acquire a lower solution viscosity to improve the PTT/GNS solution fluidity for the stable manipulation of the electrified jet. When the PTT content was decreased to 11 wt % in the PTT/GNS/TFA solution, non-Newtonian flow behavior still occurred during measurement of the shear rate, and *η* became 1/3-fold lower at 8.4 1/s (Appendix A). The *η* value of the 11 wt % PTT/GNS 90/10 solution was in the same order as that of the 7 wt % PTT/GNS 93/7 solution. Thus, reducing the PTT content in a PTT/GNS solution was found to be a good route to manipulate the stable cone-jet mode when PTT/GNS composite fibers are electrospun for high GNS content. We selected an 11 wt % PTT solution filled with 10 and 12 wt % GNS, a 9 wt % PTT solution filled with 14 wt % GNS, and a 7 wt % PTT solution filled with 16 wt % GNS to ensure a completely stable cone-jet formation during electrospinning (Table 1).

Appendix A shows the SEM and TEM images of the fiber collected from electrospinning the 11 wt % PTT solutions with 10 and 12 wt % GNS, the 9 wt % PTT solution filled with 14 wt % GNS, and the 7 wt % PTT solution filled with 16 wt % GNS. The irregular PTT composite fiber structure increased in size along the fiber. When the GNS content in the PTT solution exceeded 12 wt %, the GNS protrusion size on the electrospun fiber increased. GNS aggregation particles higher than 10 μm were observable for the PTT/GNS 86/14 and 84/16 fibers. Moreover, the nanowebs composed of nanofibrils could be observed from the electrospun PTT/GNS composite fibers, such as PTT/GNS 88/12 fibers (Appendix A) (the dashed circles indicate the nanowebs). The TEM images (Appendix A) show that the GNS aggregation particles are identifiable. When the GNS content in the PTT fiber was increased to 12 wt %, the GNS aggregation particles exceeded 2 μm and started darkening. Moreover, when the GNS content in the PTT fiber exceeded 12 wt %, the GNS aggregation particles darkened and increased in size. This observation indicates that the electrons could not pass through the GNS aggregation particles because the GNS aggregation particles were too thick. This result revealed that the GNS was layered in the PTT fiber. The GNS dispersed well in the PTT composite powders; however, the prepared PTT/GNS solution with extremely high GNS content still could not prevent GNS aggregation due to the GNS–GNS interaction at a high GNS content. The formation of PTT nanofibril is interesting. One of the hypotheses for nanofibril formation could be attributed to the fast phase separation of charged droplets generated during electrospinning [47]. However, this issue is beyond the scope of this study and deserves future work.

Based on the SEM and TEM images of the PTT/GNS composite fibers, four modes of arrangements for the GNSs in the PTT fiber were assumed upon increasing the GNSs (Figure 6). First, PTT fibers filled with 1–3 wt % GNS content were individually dispersed in the PTT fiber at intervals. Second, for the PTT fibers filled with 5–10 wt % GNS content, the GNSs were close together in the PTT fiber. Third, for PTT fibers filled with 12 wt % GNS content, some parts of the GNS overlapped in the PTT fibers. Fourth, for PTT fibers filled with 14–16 wt % GNS content, the GNS was layered in the PTT fibers. As the GNS content increased, the arrangement changed from the first arrangement to the fourth. The GNS–GNS network was apparently not formed along the PTT/GNS fiber. Therefore, the GNSs in the composite fibers were arranged differently from those in the composites, which was proposed by Zhao et al. [48].

### 3.3. Cold Crystallization of Electrospun PTT/GNS Composite Fibers

Figure 7a shows the WAXD intensity profiles of the as-spun PTT/GNS composite fiber samples. Notably, the diffraction peak at 26°, which was associated with the d-spacing of the graphitic structure, was not observed in the as-spun PTT/GNS composite fibers. According to our previous studies [29], the WAXD intensity profiles of the amorphous PTT showed a halo at 2*θ* = 20.3° and full width at half-maximum=10.8 and the characteristic diffraction peaks of the PTT crystals should be observed at 2*θ* of 15.3°, 16.8°, 19.4°, 21.8°, 23.6°, 24.6°, and 27.3° for the PTT samples. However, the WAXD intensity profiles of the as-spun neat PTT fibers showed the diffraction humps at 15°–18° and 22°–27°. This outcome was in contrast to the behavior of the PTT amorphous characteristics. The order structure is associated with the mesomorphic phase, which ws reported for neat PTT and PTT/GNS composites cold crystallization [29,49]. Therefore, the mesomorphic phase was formed during PTT electrospinning. Furthermore, when the GNS content was increased to 3 wt %, the three diffraction peaks of 16.4°, 20.0°, and 24.0° were observed at 2*θ*. The mesomorphic phase was significant during the PTT electrospinning process after the addition of GNS. Thus, GNSs can help to induce the order structure formation of the PTT chains. Given such results, the changes in GNS-filled PTT chain conformation in fiber morphologies during the subsequent cold crystallization should be further analyzed.

Figure 7b shows the DSC heating traces of the as-spun PTT/GNS composite fibers. *T_g_* was determined from the mid-point of the heat capacity jump, whereas the peak temperature of the cold crystallization was denoted as *T_c_*. Crystallization enthalpy, which was determined from the integral area of the exotherm, was represented by *ΔH_c_*. To represent the crystallizability of the PTT fiber in the presence of GNSs, *ΔH_c_* was normalized with the PTT content. The data listed in Table 2 show that the *T_g_* of PTT/GNS 100/0, 99/1, and 97/3 fibers are 40.4, 43.4, and 39.3 °C, respectively. *T_g_* remained unchanged at approximately 41.0 °C with increased GNS content. *T_c_* slightly shifted to a low temperature with increased GNS content. When the GNS content increased to 5 wt %, the exothermic peak of the PTT cold crystallization could not be clearly observed. The normalized *ΔH_c_* decreased with increased GNS content because some mesomorphic phases already developed in the as-spun PTT/GNS composite fibers prior to heating. Moreover, the melting temperature of the cold-crystallized PTT/GNS composite fibers was relatively unchanged, as shown in Figure 7b.

Figure 8 shows the FTIR spectra of the neat PTT and PTT/GNS 99/1 composite fibers during stepwise heating to 225 °C. According to literature [50,51], the absorbance bands at 875 and 870 cm^−1^ are associated with the B_3u_–CH out-of-plane bending of phenylene rings in the amorphous and crystalline phases, respectively. By contrast, the absorbance bands at 948 and 935 cm^−1^ are related to CH_2_ rocking in a gauche conformation in the crystalline phase. In the current work, the absorbance band, resulting from the trans-conformation was located at 815 and 978 cm^−1^. Prior to heating, the neat PTT and PTT/GNS 99/1 composite fibers were in the mesomorphic phase, as indicated by the intensity of the WAXD (Figure 7a). Upon heating, the neat PTT and PTT/GNS 99/1 composite fibers were gradually crystallized, as shown by the gradual absorbance increase in the 870, 935, and 948 cm^−1^ bands and the gradual absorbance decrease in the 815 and 978 cm^−1^ bands. These absorbance changes, which are consistent with previous findings on isothermal cold crystallization [49], revealed the trans-to-gauche transition behavior in PTT cold crystallization [52].

Figure 9a shows the normalized area of the 948 and 935 cm^−1^ bands and the variation in the absorbance peak at 875–870 cm^−1^ as a function of temperature for the neat PTT and PTT/GNS 99/1 composite fibers. The normalized area of the bands at 935 and 948 cm^−1^ was used in the following equation [29]:(2)$$A_{935+948}=\frac{A_{T}-A_{35}}{A_{max}-A_{35}}$$
where *A*_935+948_ is the relative crystallinity of PTT, and *A_max_* is the maximum area of the 935 and 948 cm^−1^ bands. *A_T_* and *A*_35_ are the areas of the 935 and 948 cm^−1^ bands at different temperatures and at 35 °C, respectively. The cold crystallization process was characterized by the increased *A*_935+948_ (Figure 9a), which was accompanied by a gradual band shift from 874 to 870 cm^−1^ (Figure 9b). Furthermore, a band shift was observed during heating at ~870 cm^−1^ and prior to crystal melting at approximately 200 °C, at which spectral recording became infeasible owing to the sample flow. Notably, the *A*_935+948_ value of the PTT/GNS 99/1 composite fibers at 45–50 °C was lower than that at 35 °C, and then significantly increased above 55 °C. This result indicates that the gauche conformation of the PTT/GNS composite fibers decreased at temperatures higher than *T_g_*. The gauche conformation content of the PTT chains for PTT/GNS composite fibers could be higher than that for neat PTT fibers during the solution jet and bending instability in the electrostatic field due to the presence of GNSs, but the mild increase in regularity is not sufficient to cause the formation of crystalline structures. Thus, the chain structure of PTT was rearranged again after subsequent heating treatments. The *A*_935+948_ of the PTT/GNS 99/1 composite fibers at 65 °C was higher than that of the neat PTT fiber at 65 °C. The saturated *A*_935+948_ of the PTT/GNS 99/1 composite fibers at 80 °C was also higher than that of the neat PTT fiber at 90 °C. Meanwhile, the band between 875 and 870 cm^−1^ of the PTT/GNS 99/1 composite fibers was slightly lower than that of the neat PTT fiber at a high-temperature region (Figure 9b). These results indicate that the cold crystallization of the PTT composite fibers was enhanced by the GNSs. Consistent results were obtained between DSC and FTIR measurements with regard to the kinetics of cold crystallization in PTT fibers.

Figure 10a shows the WAXD of the neat PTT fibers during stepwise heating to 240 °C. The WAXD intensity profiles of as-spun neat PTT fibers showed the mesomorphic phase characteristics prior to heating. When the temperature was increased to 60 °C, the intensity of diffraction humps at 15.0°–18.0° and 22.5°–25.5° began to increase. When the temperature was increased to 120 °C, the characteristic diffraction peaks at 15.6° and 17.2° appeared. When the temperature was increased to 170 °C, the characteristic diffraction peaks of the PTT crystals were clearly observed and increased gradually at the 2*θ* of 15.6°, 17.2°, 19.4°, 21.4°, 23.2°, and 24.5°. Figure 10b shows the WAXD of the PTT/GNS 99/1 composites fibers during stepwise heating to 240 °C. A similar increase in the WAXD intensity profiles was observed in the PTT/GNS 99/1 composite fibers with the increased temperature. Therefore, neat PTT and PTT/GNS 99/1 composite fibers crystallizes from a mesomorphic phase into a triclinic structure at 170 °C during cold crystallization. The neat PTT and PTT/GNS 99/1 composite fibers were still mesomorphic phases during *A*_935+948_ increase in the FTIR spectra and exothermic peak in DSC, which is associated with the PTT cold crystallization kinetics.

Based on our previous studies [29], the PTT lamellae could be epitaxially grown on top of the GNS. Furthermore, the mesomorphic phase was formed during PTT electrospinning, and GNSs could induce the PTT mesomorphic phase significantly during PTT/GNS electrospinning in this study. During cold crystallization, the PTT mesomorphic phase could be the PTT nuclei, which were randomly developed on the GNS surface, to form the lamellae in PTT/GNS composite fiber (Figure 11). The PTT cold crystallization rate in the PTT/GNS composite fiber increased with the gradual addition of GNSs. The enhanced cold crystallization kinetics was attributed to the high nucleation ability of GNSs, which results from PTT mesomorphic phase on their surfaces, while the overgrown lamellae from the GNS surface of the PTT/GNS composite fibers may result in the retardation alignment of PTT crystalline lamellae in the PTT/GNS composite fibers under quiescent heating. Therefore, the orientation of PTT lamellae in the PTT fibers is higher than that in the PTT/GNS composite fibers.

### 3.4. Mechanical Properties of Electrospun PTT/GNS Composite Fibers

Based on cold crystallization results, the PTT/GNS composite fibers crystallizes into a triclinic structure at 170 °C. Therefore, 170 °C was selected for annealing. Figure 12 shows the stress-strain curves of PTT/GNS CFMs with and without annealing at 170 °C. The determined Young’s modulus (*E*), tensile strength (*σ_max_*), elongation at break (*ε_max_*), porosity, and *d_f_* are displayed in Table 3. The as-spun neat PTT fiber mats prior to 170 °C annealing possessed higher *E*, larger *σ_max_*, and longer *ε_max_* than after 170 °C annealing. Similar results were obtained in the PTT/GNS CFMs. Annealing could lead to an increase in the porosity and *d_f_*. Moreover, the *E* and *σ_max_* of the PTT/GNS CFMs were decreased, and the *ε_max_* of the PTT/GNS CFMs were increased with increased GNS content, regardless of annealing. Based on the cold crystallization and solution viscosity results of the as-spun PTT/GNS CFMs, the present mechanical reduction is attributed to the alignment of the PTT chains in the PTT/GNS composite fibers during electrospinning. When the viscosities of the PTT/GNS solution were increased with the increase in GNS content, the *d_f_* of the electrospun PTT/GNS composite fibers was decreased first and then increased with increased GNS content. This phenomenon resulted in retarded alignment of the PTT chains in the PTT/GNS composite fibers during electrospinning. Therefore, the PTT/GNS CFMs became ductile with the addition of GNSs. In addition, the PTT chains in the PTT/GNS composite fibers relaxed via thermal expansion during thermal heating when the temperature exceed the *T_g_* of PTT. Thus, the *d_f_* values of the PTT/GNS composite fibers at 170 °C annealing temperature was slightly larger than those of the as-spun PTT/GNS composite fibers. The alignment of the PTT crystalline lamellae in the PTT/GNS composite fibers was inhibited by the PTT nuclei on the GNS surface after annealing at this temperature (Figure 11).

Chen et al. [24] found that the PTT filaments have a high instantaneous elastic recovery even at a high elongation of 20%, whereas the clothing materials, such as filaments, are usually used at elongations below 20%. Therefore, we selected 20% elongation for the instantaneous elastic recovery measurement. Appendix A show the stress-strain curves of PTT/GNS CFMs filled with various amounts of GNS in five stretch cycles by the 20% elongation limit with and without annealing at 170 °C. The elastic recovery results are listed in Table 3. When the crosshead returned to its original position, no PTT/GNS CFMs could return to their original position, and the remnant elongation slightly increased with increasing cycles. The elastic recoveries of the PTT/GNS CFMs with 170 °C annealing are better than those of the as-spun PTT/GNS CFMs. The elastic recoveries of the PTT/GNS CFMs were slightly reduced with increased GNS content. The WAXD and FTIR results indicate that the as-spun PTT/GNS CFMs are of the mesomorphic phase, and the PTT/GNS CFMs with 170 °C annealing have a triclinic crystal structure with numerous gauche conformations. This results indicates that the elastic recoveries of the PTT/GNS CFMs are mainly caused by the gauche conformations of the PTT crystals in the fiber.

### 3.5. Electrical Properties of Electrospun PTT Composite Fiber Mats

Figure 13a shows the *σ* of PTT composites and PTT fibers filled with GNS. When the GNS contents were increased, the GNS gradually formed a conductive path. The percolation scaling laws are considered suitable in revealing the *σ* of the PTT/GNS CFMs. Thus, in this section, the addition of GNS to PTT/GNS CFMs have to be presented in the volume percentage on the basis of the percolation equation. For the PTT/GNS composite fibers, the PTT/GNS CFMs exhibited an improvement in conductivity, and a gradual transition was observed from 1.77 to 9.98 vol % (3–16 wt %). The *σ* of the PTT/GNS CFMs became 3.18 × 10^−5^ S/cm following a GNS content of 9.98 vol %. Compared with that of the neat PTT fibers, a 9-order increase in *σ* was observed for the PTT/GNS CFMs, with 7.35 vol % GNS content. However, compared with our previous study [21], wherein GNSs were used in PTT/GNS composites using the same GNSs as this study, the PTT/GNS CFMs in the present study exhibited significantly lower conductivity than that of PTT/GNS composites with the same GNS content. PTT/GNS composites could form a conductive network more easily than PTT/GNS CFMs. This discrepancy could be attributed to the highly porous electrospun fiber mats with voids filled with insulating air. Chien et al. [23] demonstrated that the conductivity of electrospun PDLLA/CNC CFMs may be affected by the porosity of CFMs. Therefore, we prepared a PTT/GNS composite film via hot pressing PTT/GNS CFMs (HP-PTT/GNS CFMs) at 280 °C. The *σ* of the HP-PTT/GNS CFMs was similar to that of the PTT/GNS composites that were prepared from hot-pressed dried powders. The porous structure of the PTT/GNS CFMs is one of the major reasons causing the difference in conductive behavior between the PTT/GNS composites and the PTT/GNS CFMs. The hot-pressed fiber mats eliminated the insulating voids.

To determine quantitatively the minimum volume fraction of GNS required for developing the network for electron transportation, the percolation scaling law was applied to describe the relationship between *σ* and ϕ − ϕ*_c_*. Corresponding plots are provided in Figure 13b. ϕ*_c_* and *β* values were determined based on the regression analysis. The experimental ϕ*_c_* values of the PTT/GNS CFMs is 6.10 vol %. The derived ϕ*_c_* of the PTT/GNS CFMs is higher than that of the PTT/GNS composites (0.84 vol %) [21]. The derived *β* values of the PTT/GNS CFMs is 6.52, which is substantially different from the universal value of 2.0 for a three-dimensional lattice. In addition, several studies analyzed similar polymer/carbon nanofiller CFM samples. For polymer/carbon nanofiller CFMs, the *β* of the PDLLA/CNC CFMs is 3.20 [23], while one of the PS/CNT CFMs is 0.80 [22]. The *β* of the PTT/GNS CFMs is significantly higher than those of the PDLLA/CNC and PS/CNT CFMs. These discrepancies could be attributed to the route of dispersion state of the fillers, their geometrical structures, the intrinsic differences in the polymer/GNS pair, or the porosity of the electrospun fiber mats. Nevertheless, the *β* values of the PTT/GNS CFMs are significantly higher than those of the PTT/GNS composites (3.91) [21]. However, this result is different from the percolation exponent result for the PDLLA/CNC composite fiber mats [23]. The *β* of the PDLLA/CNC CFMs is similar to that of the PDLLA/CNC casting composite film. This phenomenon implies that the conductivity of the PTT/GNS fiber mats is not merely affected by the porosity of the electrospun fiber mats.

*L_GNS_* in the PTT/GNS/TFA solution was significantly decreased with 5 wt % GNS content due to the significant increase of *G′* and solution conductivity (Figure 3a and Appendix A), whereas the conductivity of the electrospun PTT/GNS 95/5 CFMs (8.90 × 10^−16^ S/cm) is significantly low. Moreover, the value of the GNS–GNS network formation in the PTT/GNS CFMs was larger than those in the PTT/GNS composites and the PTT/GNS/TFA solution. These results imply that the GNS–GNS network in the solution did not form a GNS–GNS conductive network along the PTT/GNS fiber during electrospinning. However, a GNS conductive fiber could be successfully made using another spinning method. Jalili et al. [53] showed that GO fibers could be produced at a low GO content (0.075 wt %, the GO liquid crystalline network) via wet-spinning, and that a GO–GO network would be formed in the GO fibers. The difference of the filler network formation between GO fibers and PTT/GNS fibers is caused by the processing method. The wet-spinning process could put GO in order along the GO fiber because the jet stretch ratio (stretching velocity/injection velocity of the flow) could be low (0.5–1.5) during wet-spinning [54]. The electrospun PTT/GNS fiber process is illustrated in Figure 13c. Wang et al. [32] found that the drawdown ratio could be estimated at approximately 27,000 from the Taylor cone apex up to the straight jet end (spinneret diameter/jet-end diameter)^2^. The drawdown ratio was approximately 20–80 at the whipping region (jet-end diameter/fiber diameter)^2^. When the GNS–GNS network in the composite solution flowed from the Taylor cone apex into a straight jet, *L_GNS_* was stretched in the straight jet and was subsequently stretched again at the whipping region under an electric field. Therefore, *L_GNS_* could be increased; the GNS–GNS network would not be formed along the PTT/GNS composite fibers despite the electrospun PTT/GNS composite fibers from the GNS–GNS network solution.

When *L_GNS_* was increased via electrostatic stress, the electrons could not be transported along the PTT/GNS composite fiber. However, the lateral dimension of GNS was larger than the diameter of the electrospun PTT fibers. Hence, GNS aggregated particles protruded from the smooth PTT fiber. The protruding parts of GNS in the PTT/GNS composite fibers and in the neighboring PTT/GNS fibers helps in the formation of the GNS–GNS network. The electrons were transported in the GNS-GNS network that was formed via GNS aggregations in the neighboring PTT/GNS fibers with a non-woven structure. (Figure 13d) The percolated GNS–GNS networks in the PTT/GNS CFMs are different from those in the PTT/GNS composite. Several studies [21,55,56,57] have indicated that when conduction-insulation composites experience a tunneling effect between conducting particles, the percolation exponent will be in the range of *β_o_* and *β_o_*_+10_, where *β_o_* is the universal conductivity exponent. The percolation exponent value of the PTT/GNS CFMs was higher than that of the PTT/GNS composites because the porous structures and GNS–GNS network of the PTT/GNS CFMs were increased in *L_GNS_*, thereby promoting a tunneling effect [58] (Figure 13d).

## 4. Conclusions

PTT composite powders filled with well-dispersed GNSs were prepared via coagulation, and the composite powders were re-dissolved in a TFA solvent to obtain well-dispersed PTT/GNS solutions. The effects of increased GNS concentration on the electrospinning solution and process, as well as the morphologies and property variations of PTT/GNS composite fibers, were investigated using several analytical techniques, including rheometers, conductivity meters, SEM, and TEM. The addition of GNS increased the PTT solution viscosity and conductivity. After electrospinning the 14 wt % PTT solution with various GNS contents, the *d_f_* of the PTT/GNS fibers initially decreased and then increased as the GNS was added into the PTT solution. The decreased *d_f_* and the increased *d_f_* of the PTT/GNS fiber were initially dominated by the solution conductivity, and subsequently by the solution viscosity, respectively. The morphologies of the GNS dispersion in the PTT composite fiber changed as the GNS content increased. Moreover, GNSs were embedded and protruded from the fibers. Regardless of GNS content, a PTT mesomorphic phase was formed during electrospinning, whereas GNSs could induce PTT mesomorphic phase significantly during PTT/GNS electrospinning. The PTT cold crystallization rate of PTT/GNS composite fibers increased with the GNS content. During cold crystallization, the PTT mesomorphic phase could be the PTT nuclei, which were randomly developed on the GNS surface, to form the lamellae in the PTT/GNS composite fiber. The increased viscosities of PTT/GNS solution retarded the alignment of PTT chains in the PTT/GNS composite fibers during the electrospinning process, whereas the alignment of PTT crystalline lamellae in the PTT/GNS composite fibers was inhibited by the PTT nuclei on the GNS surface after annealing at 170 °C. Therefore, the mechanical properties of PTT/GNS CFMs became ductile with the addition of GNSs. The elastic recoveries of the PTT/GNS CFMs with 170 °C annealing are better than those of the as-spun PTT/GNS CFMs. The elastic recoveries of the PTT/GNS CFMs were slightly reduced with increased GNS content. The PTT/GNS composites could form a conductive network more easily than PTT/GNS CFMs. The conductivity thresholds of the PTT/GNS CFMs are higher than that of the PTT/GNS composite films because the electrospun fiber mats were highly porous and contained voids filled with insulating air, and the GNS–GNS network could not be formed along the composite fibers. The porous structure and GNS dispersion of the elicited PTT/GNS CFMs are important to the CFMs’ performance, especially for *σ*. The *σ* of the HP-PTT/GNS CFMs were similar to those of the PTT/GNS composites that were prepared from hot-pressed dried powders.

## Figures and Tables

**Figure 1 polymers-11-00164-f001:**
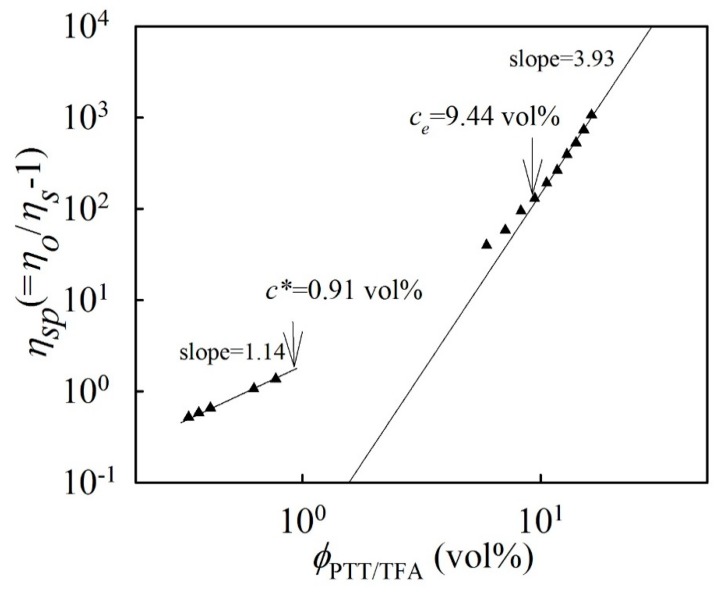
Concentration dependence of the specific viscosity of the PTT solutions.

**Figure 2 polymers-11-00164-f002:**
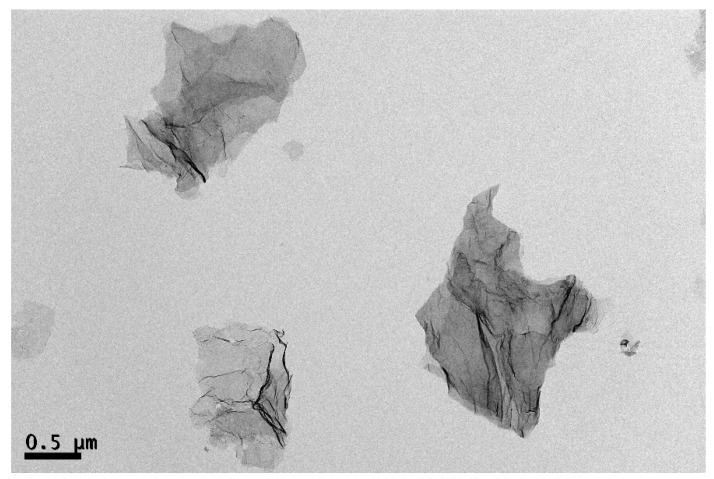
Transmission electron microscope (TEM) images of GNS the deposited GNS on the TEM grid prepared from the ortho-dichlorobenzene (*o*-DCB) solution.

**Figure 3 polymers-11-00164-f003:**
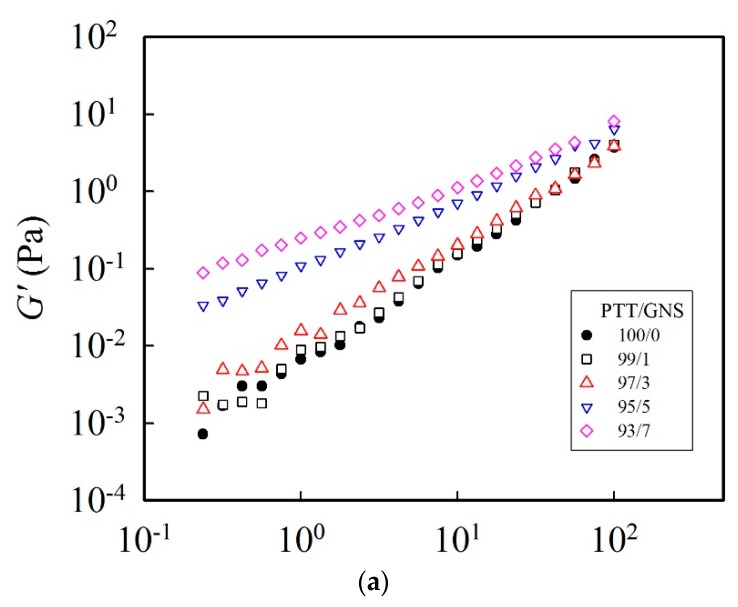
Effect of GNS content on the viscoelastic properties of PTT solutions: (**a**) dynamic storage modulus *G′*, (**b**) dynamic loss modulus *G″*, and (**c**) complex viscosity *η** at 25 °C. The PTT/trifluoroacetic acid (TFA) concentration is 14 wt %.

**Figure 4 polymers-11-00164-f004:**
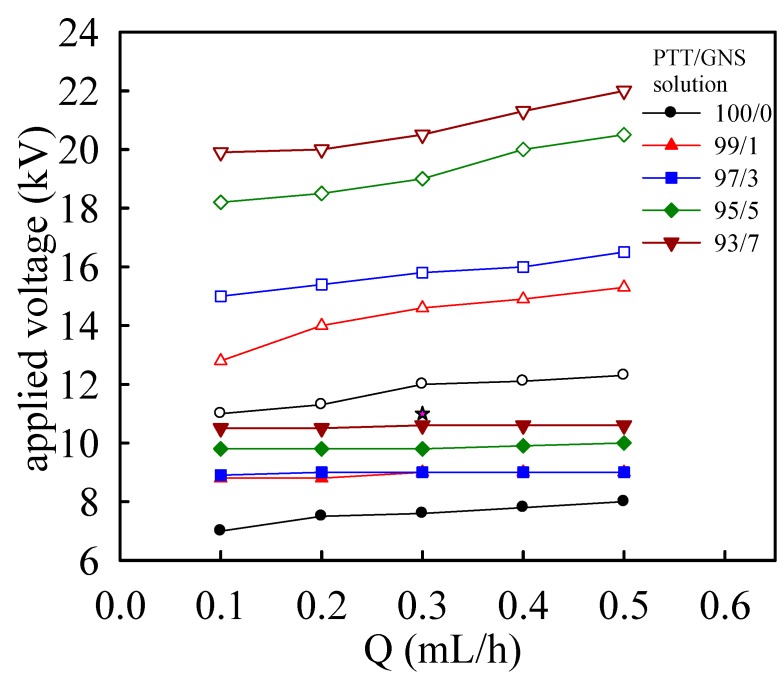
Functional domain for electrospinning of 14 wt % PTT solution with various GNS contents. The domains indicate the range of operating electrical fields required for the stable cone-jet mode. (Filled symbols for lower bond applied voltage and open symbols for upper bond applied voltage).

**Figure 5 polymers-11-00164-f005:**
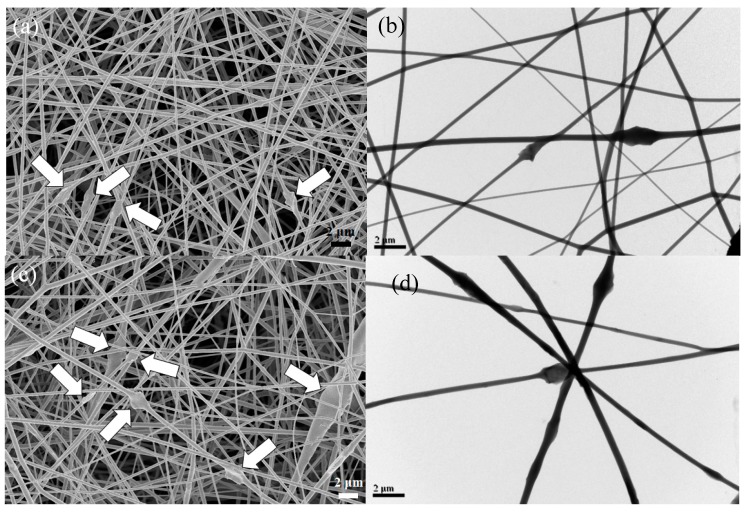
Scanning electron microscope (SEM) and TEM images of electrospun PTT fibers filled with: (**a**,**b**) 1 wt %, (**c**,**d**) 3 wt %, (**e**,**f**) 5 wt %, and (**g**,**h**) 7 wt % GNS. The positions of GNSs and nanofibrils are indicated by the thick and thin arrows, respectively.

**Figure 6 polymers-11-00164-f006:**
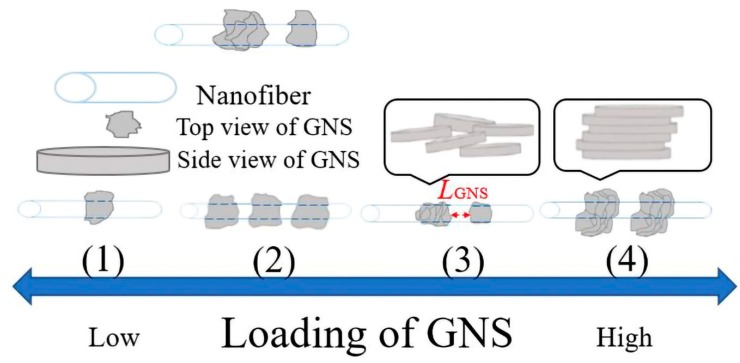
Schematic models of various dispersion types of GNSs in PTT fiber: (**1**) GNSs are individually dispersed in the PTT fiber at intervals. (**2**) GNSs are close together in the PTT fiber. (**3**) Some parts of the GNSs overlap with one another in the PTT fiber. *L_GNS_* indicates the inter-GNS distance assemblies in the PTT composite fiber. (**4**) The GNS are layered in the PTT fiber. When the GNS content is increased, the resulting dispersion form gradually changes.

**Figure 7 polymers-11-00164-f007:**
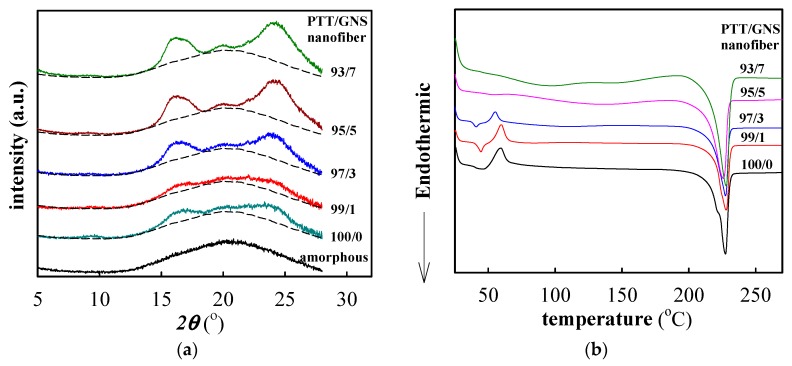
(**a**) Wide-angle X-ray diffraction (WAXD) intensity profiles of the as-spun PTT/GNS composite fibers, and (**b**) differential scanning calorimetry (DSC) heating traces of the as-spun PTT/GNS composite fibers.

**Figure 8 polymers-11-00164-f008:**
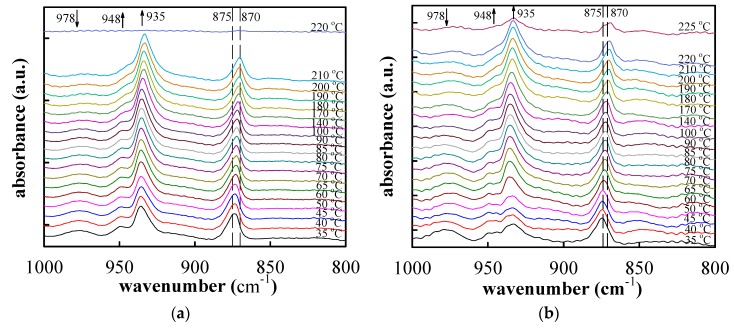
Fourier transform infrared spectroscopy (FTIR) spectra of (**a**) neat PTT and (**b**) PTT/GNS 99/1 composite fibers during stepwise heating to 225 °C.

**Figure 9 polymers-11-00164-f009:**
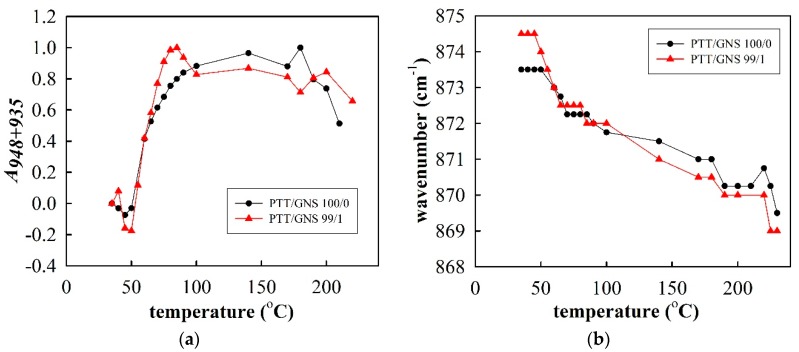
(**a**) Normalized area of the 948 and 935 cm^−1^ bands (*A*_935+948_) and (**b**) variation in the absorbance peak at 875–870 cm^−1^ for the neat PTT and PTT/GNS 99/1 composite fibers during stepwise heating.

**Figure 10 polymers-11-00164-f010:**
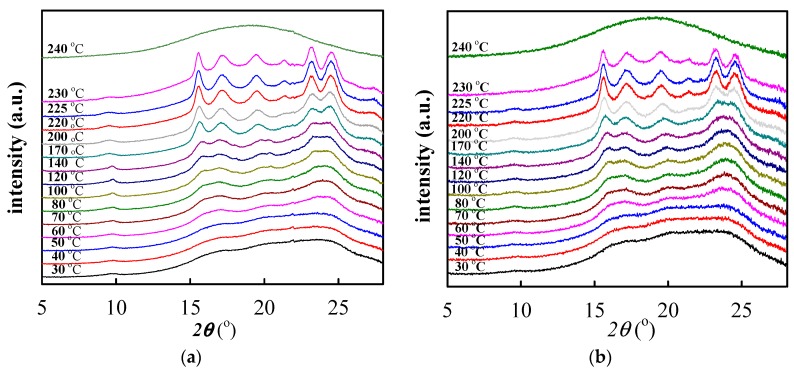
WAXD intensity profiles of (**a**) neat PTT and (**b**) PTT/GNS 99/1 composite fibers during stepwise heating to 240 °C.

**Figure 11 polymers-11-00164-f011:**
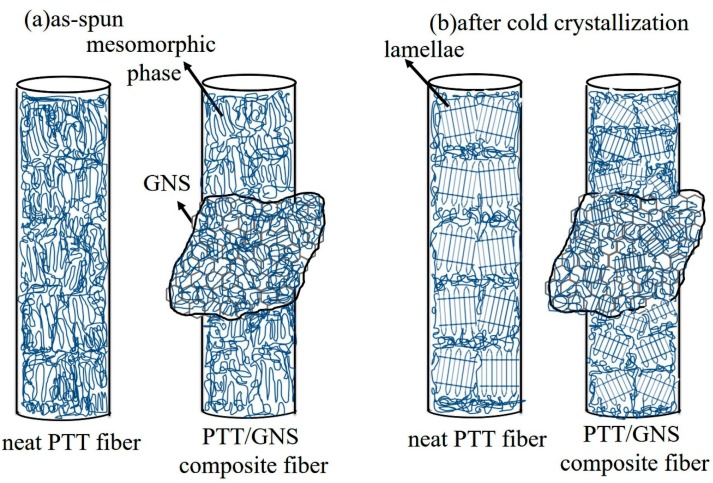
Schematic of the microstructure change of the PTT/GNS composite fibers during cold crystallization.

**Figure 12 polymers-11-00164-f012:**
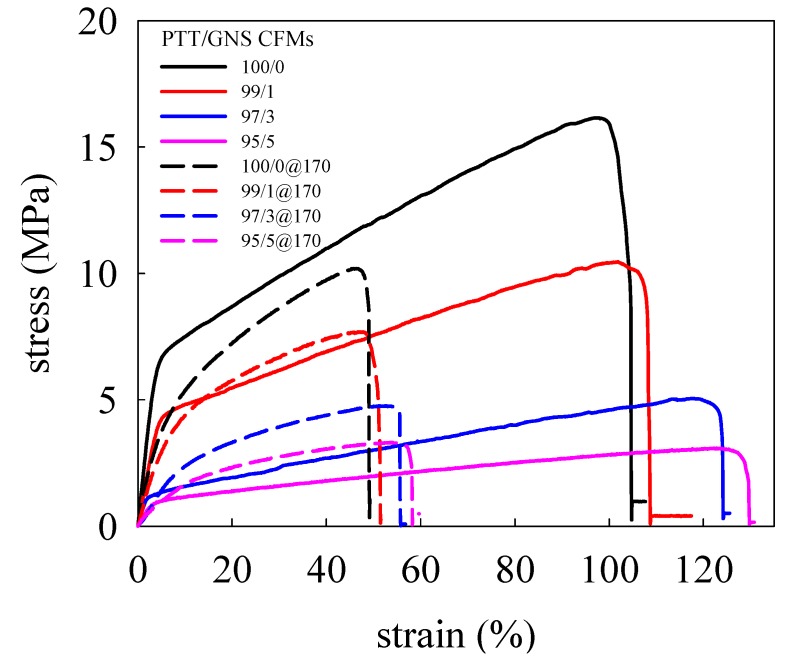
Stress-strain curves of PTT/GNS CFMs with and without annealing at 170 °C for 30 min. The solid lines indicate the as-spun PTT/GNS CFMs, and the dashed lines indicate PTT/GNS CFMs with annealing at 170 °C.

**Figure 13 polymers-11-00164-f013:**
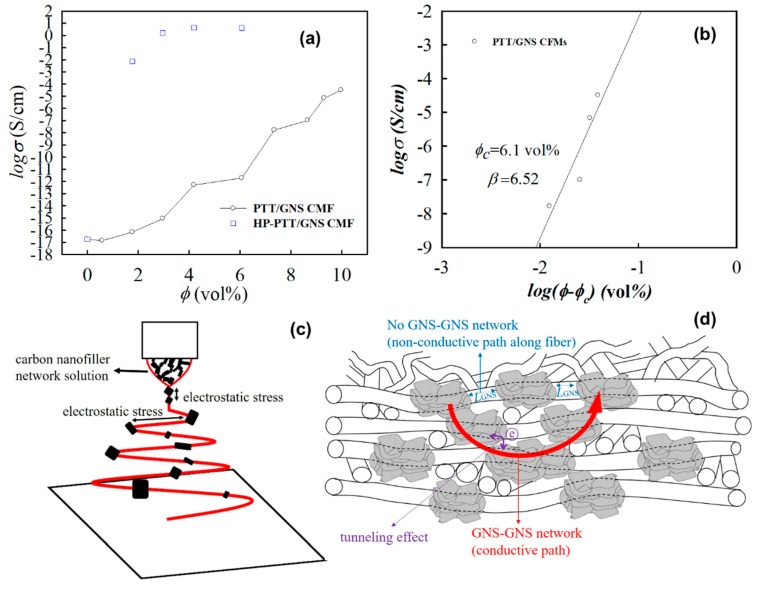
(**a**) Conductivity versus GNS volume contents of PTT/GNS CFMs and HP-PTT/GNS CFM composites. (**b**) Percolation scaling law between *σ* and ϕ − ϕ*_c_* for PTT/GNS CFMs. (**c**) Schematic representation of electrospun PTT/GNS fiber. (**d**) Schematic representation of GNS-GNS conductive path in the PTT/GNS CFMs.

**Table 1 polymers-11-00164-t001:** Summary of composite fibers obtained and the resulting fibers morphologies.

Poly(trimethylene terephthalate) (PTT) Concentration (wt %)	Graphene Nanosheet (GNS) Concentration (wt %)	Fiber Morphology
7, 8	-	Beads on fiber
9, 11, 12, 14	-	Smooth fiber with nanofibril
14	1, 3, 5, 7	Irregular fiber structure with nanofibril; irregular fiber structure depend on GNS concentrations
11	10, 12	Irregular fiber structure with nanofibril; irregular fiber structure size depend on GNS concentrations
9	14	Irregular fiber structure with nanofibril and big GNS aggregation particles
7	16	Irregular fiber structure with nanofibril and big GNS aggregation particles

**Table 2 polymers-11-00164-t002:** Thermal properties of the as-spun PTT/GNS composite fibers.

PTT/GNS fibers	*T_g_* (°C)	*T_c_* (°C)	*ΔH_c_* (J/g)	*T_m_* (°C)	*ΔH_m_* (J/g)
100/0	40.4	59.4	10.1	227.4	66.21
99/1	43.4	59.7	9.9	227.9	41.20
97/3	39.3	55.3	5.6	227.4	46.95
95/5	-	-	-	225.8	48.51
93/7	-	-	-	225.9	43.24

**Table 3 polymers-11-00164-t003:** Mechanical properties of PTT/GNS CFMs with and without annealing at 170 °C for 30 min.

PTT/GNS CFMs	*E* (MPa)	σ*_max_* (MPa)	*ε_max_* (%)	*d_f_* (nm)	Porosity (%)	Fifth Elastic Recovery (%)
100/0	141.6 ±30.7	15.1 ± 2.0	102.5 ±7.7	256 ± 92	46	85.8
99/1	89.4 ± 5.0	10.0 ± 0.5	106.9 ±2.8	141 ± 28	57	84.4
97/3	52.9 ± 11.2	4.4 ± 0.5	109.8 ± 8.7	310 ± 116	43	83.4
95/5	22.6 ± 7.2	3.0 ± 0.1	123.8 ± 3.6	359 ± 133	39	83.9
100/0@170	102.0 ± 6.5	9.9 ± 0.4	48.6 ± 3.5	290 ± 87	47	91.8
99/1@170	77.4 ± 5.4	7.4 ± 0.2	49.6 ± 1.9	237 ± 81	59	91.7
97/3@170	37.0 ± 3.0	4.9 ± 0.2	53.7 ±1.8	347 ± 128	43	90.9
95/5@170	35.0 ± 3.9	3.5 ± 0.1	56.9 ± 3.3	366 ± 134	43	90.4

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
