# Peer review of "Electrospun Graphene Nanosheet-Filled Poly(Trimethylene Terephthalate) Composite Fibers: Effects of the Graphene Nanosheet Content on Morphologies, Electrical Conductivity, Crystallization Behavior, and Mechanical Properties"

_polymers, 2019, doi:10.3390/polym11010164_

Reviewer 1 Report

In this manuscript, the authors prepared PTT/GNS composite fibers with electrospinning and then investigated the effects of GNS content on the morphologies, conductivity, crystallization behavior, and mechanism properties of the fabricated composite fibers. Based on the experimental characterizations, the formation mechanism, electrical conductivity mechanism, and crystallization behavior of the PTT/GNS composites were proposed correspondingly. It is a very interesting and comprehensive study on the properties of electrospun graphene-polymer fiber materials. The experiments are good-designed, and the manuscript is well-written. All the conclusions are supported by the presented data. Therefore, this manuscript is recommended for publication at Polymers after minor revisions.

Special comments for the revision:

The “Introduction” part is too long. It is suggested for the authors to delete some introduction on the case studies (such as ** et al. demonstrated **). In addition, a few related references (such as Carbon 2012, 50, 5605; J Mater Chem B 2017, 5, 1699; Adv Funct Mater, 2016, 26, 2122.) could be considered to add. The authors are suggested to add a few sentences to indicate the novelty and significance of this work.

In Part 2, it is necessary for the authors to add another section on “Characterization techniques”, in which the details on the characterizations of DSC, FTIR, WAXD, and others could be introduced.

How did the authors synthesize the GNS? With which method? The GNS is GO or reduced GO? What is the size of GNS?

Why the small amount of GNS (1%) reduced the diameter of fibers? Meanwhile the higher amount increased the diameter of composite fibers? More discussion is needed.

In Figure 4, the authors utilized SEM and TEM to observe the electrospun PTT/GNS composite fibers. How to prove the filling of GNS into the PTT fibers? Is it possible to measure GNS in the composite fibers by FTIR, WAXD, or even Raman techniques? More discussion is needed.

Author Response

Answers to the comments from Reviewer#1:

1. The “Introduction” part is too long. It is suggested for the authors to delete some introduction on the case studies (such as ** et al. demonstrated **). In addition, a few related references (such as Carbon 2012, 50, 5605; J Mater Chem B 2017, 5, 1699; Adv Funct Mater, 2016, 26, 2122.) could be considered to add. The authors are suggested to add a few sentences to indicate the novelty and significance of this work.

Response: The Introduction section has been rewritten, and related literature has been cited. The changes in the section are reflected on page 1-3 of the revised manuscript (red text).

2. In Part 2, it is necessary for the authors to add another section on “Characterization techniques”, in which the details on the characterizations of DSC, FTIR, WAXD, and others could be introduced.

Response: The Materials and Methods section has been rewritten and additional details (FTIR, WAXD, and DSC) have been provided on page 4 (line 186-195) of the revised manuscript (red text).

3. How did the authors synthesize the GNS? With which method? The GNS is GO or reduced GO? What is the size of GNS?

Response: The additional details of GNS have been provided on page 3 (line 141-143) of the revised manuscript (red text).

4. Why the small amount of GNS (1%) reduced the diameter of fibers? Meanwhile the higher amount increased the diameter of composite fibers? More discussion is needed.

Response: Additional discussion on the diameter of PTT/GNS composite fibers has been provided on page 10 (line 364-368) of the revised manuscript (red text).

5. In Figure 4, the authors utilized SEM and TEM to observe the electrospun PTT/GNS composite fibers. How to prove the filling of GNS into the PTT fibers? Is it possible to measure GNS in the composite fibers by FTIR, WAXD, or even Raman techniques? More discussion is needed.

Response: Owing to the chemical structure of PTT, the absorbance band of GNS for the FTIR and Raman spectra could overlap with that of PTT for the FTIR and Raman spectra. For WAXD measurement, no diffraction peaks at 2°-30° was detected because of highly exfoliated graphene. Therefore, TEM and SEM are effective in determining the location of GNS in the PTT/GNS composite fibers.

We appreciate the helpful comments and suggestions provided by the reviewer to enhance the readability of this manuscript.

Reviewer 2 Report

The paper from Huang and co-workers thoroughly investigates the effects of increased graphene nanosheet (GNS) concentration on structure and properties of the electrospun GNS-filled poly(trimethylene terephthalate) (PTT/GNS) composite fibers. The paper is well written and the conclusions are well supported by the experimental data. Minor remark: DSC experimental parameters are not described in 2. Materials and Methods and should be added.

Author Response

Answers to the comments from Reviewer#2:

1.      DSC experimental parameters are not described in 2. Materials and Methods and should be added.

Response: The Materials and Methods section has been rewritten and additional details (FTIR, WAXD, and DSC) have been provided on page 4 (line 186-195) of the revised manuscript (red text).

We appreciate the helpful comments and suggestions made by the reviewer to enhance the readability of this manuscript.

Reviewer 3 Report

The work titled “Electrospun graphene nanosheet-filled poly(trimethylene terephthalate) composite fibers: effects of the graphene nanosheet content on morphologies, electrical conductivity, crystallization behavior, and mechanical properties” describes fabrication of mechanically and electrically advanced composite materials. This work is interesting and carries many experiments demonstrating properties of poly(trimethylene terephthalate) co-mixed with graphene nanosheets (GNSs) in good TFA solvent, which has potential for applications in energy and electronic devices. Authors showed that different loading amount results in mechanical and conducting changes over 9 orders of magnitude. Despite credit of this work is high, some small questions still can be addressed.

In the introduction part, authors can mention some state-of-the art electrical properties of SWNTs, which were already reported in such works as Adv.Mater.2012, 24, 6147–6152, Adv.Mater.2013, 25, 2948–2956 and Adv.Mater.2014, 26, 5969–5975. As well as advantages of composite materials based on graphene oxide (Nature Communications volume 6, Article number: 8817 (2015)). 

Lines 336-341: Author observed difference in nanofibers diameters as different amount of GNS material was introduced into the composites. This finding is interesting; which can be used for a demonstration how conductivity changes in respect to nanowires thickness. Did authors study single nanowire conductivity or can they take in account this?

Line 385: Typo mistake in units.

Line 384: Authors discussed conductivity of their materials, which rises gradually as the wt% of GNSs concentration increases in prepared sample. Did author measured their samples before or after cold crystallization step? If so, is there some essential difference? Can authors discuss if there is any impact of lamella structure on the conductivity?

Line 649: Authors explained difference in conductivity by tunneling mechanism via GNS aggregations, which can be still very inefficient process since the volumetric ration to non-conducting polymer is very high in comparison to conducting phase and such conductivity can be explained as a percolation process, where the contact area of the contact might play a role. Did authors investigate this question in details?  Did they measure ohmic shape of I-V characteristics for each sample? Schematic picture of tunneling mechanism between conducting particles can be helpful for understanding.   

Author Response

Answers to the comments from Reviewer#3:

1. In the introduction part, authors can mention some state-of-the art electrical properties of SWNTs, which were already reported in such works as Adv.Mater.2012, 24, 6147–6152, Adv.Mater.2013, 25, 2948–2956 and Adv.Mater.2014, 26, 5969–5975. As well as advantages of composite materials based on graphene oxide (Nature Communications volume 6, Article number: 8817 (2015)).

Response: Related literatures has been cited in the Introduction section. The changes in the Introduction section are reflected on page 1 (line 31) of the revised manuscript (red text).

2. Lines 336-341: Author observed difference in nanofibers diameters as different amount of GNS material was introduced into the composites. This finding is interesting; which can be used for a demonstration how conductivity changes in respect to nanowires thickness. Did authors study single nanowire conductivity or can they take in account this?

Response: PTT is not a conducting polymer. According to schematic representation of electrospun PTT/GNS fiber (Figure 12), LGNS was large along the PTT/GNS composite fibers. Thus, the electron was not be transported along the PTT/GNS composite fiber. The conductivity of the single PTT/GNS composite fiber cannot be measured.

3. Line 385: Typo mistake in units.

Response: We appreciate the corrections on the contribution file made by the reviewer. The typos have been corrected in the revised manuscript (line 402) (red text).

4. Line 384: Authors discussed conductivity of their materials, which rises gradually as the wt% of GNSs concentration increases in prepared sample. Did author measured their samples before or after cold crystallization step? If so, is there some essential difference? Can authors discuss if there is any impact of lamella structure on the conductivity?

Response: We did not measure the conductivity of the PTT/GNS CFMs after cold crystallization. Given that the PTT mesomorphic phases already developed in the as-spun PTT/GNS composite fibers prior to heating and the difference between PTT mesomorphic phase and PTT triclinic structure was the order structure of PTT chain, the location of the GNS-GNS networks in the PTT/GNS CFMs would not change after cold crystallization. Thus, the conductivity of PTT/GNS CFMs would not change after cold crystallization.

5. Line 649: Authors explained difference in conductivity by tunneling mechanism via GNS aggregations, which can be still very inefficient process since the volumetric ration to non-conducting polymer is very high in comparison to conducting phase and such conductivity can be explained as a percolation process, where the contact area of the contact might play a role. Did authors investigate this question in details?  Did they measure ohmic shape of I-V characteristics for each sample? Schematic picture of tunneling mechanism between conducting particles can be helpful for understanding.

Response: The schematic representation (Figure 12(d)) of the tunneling effect among the GNS aggreations in the PTT/GNS CFMs are reflected on page 19 (line 623) of the revised manuscript (red text). The samples with high electrical conductivity (>10-6 S/cm) were measured with a Keithley 2400 SourceMeter. Therefore, we measured I-V characteristics of the samples with high electrical conductivity (PTT/GNS 86/14 and 84/16 CFMs). The I-V characteristics of PTT/GNS 86/14 and 84/16 CFMs are shown in below.

  (b) 

  (a) 

The I-V characteristics for PTT/GNS (a) 84/16 and (b) 86/14 CFMs

We appreciate the helpful comments and suggestions made by the reviewer to enhance the readability of this manuscript.
